# Revealing Pan-Histology Immunomodulatory Targets in Pediatric Central Nervous System Tumors

**DOI:** 10.3390/cancers15225455

**Published:** 2023-11-17

**Authors:** Robert T. Galvin, Sampreeti Jena, Danielle Maeser, Robert Gruener, R. Stephanie Huang

**Affiliations:** 1Division of Pediatric Hematology & Oncology and Bone Marrow Transplant, University of Minnesota, Minneapolis, MN 55455, USA; rgalvin@umn.edu; 2Department of Experimental and Clinical Pharmacology, University of Minnesota, Minneapolis, MN 55455, USA; jenax004@umn.edu (S.J.); rgruener@umn.edu (R.G.); 3Department of Bioinformatics and Computational Biology, University of Minnesota, Minneapolis, MN 55455, USA; maese005@umn.edu

**Keywords:** immunotherapy, pediatric CNS malignancies, tumor immunosuppression, tumor microenvironment, epigenetics

## Abstract

**Simple Summary:**

Pediatric brain tumors, comprising both benign and malignant forms, rank as the second most prevalent neoplasms in children and confer the highest mortality among pediatric cancer cases. While the immune system exhibits anticancer capabilities under certain circumstances, tumors deploy diverse mechanisms to stifle immune responses. These mechanisms are poorly understood in pediatric brain tumors, which are developmentally unique. In this study, we were motivated to find common mechanisms of immune resistance employed by aggressive pediatric brain tumors, leveraging patient-derived molecular data from the expansive Childhood Brain Tumor Network. Our analyses identify shared immunologic clusters between tumor types that correlate with patient outcomes, immune cell content, and immunosuppressive gene expression. We identify various immunosuppressive genes that contribute to worse outcomes within clusters. We further utilize this dataset to examine genes implicated in preventing immune recognition in malignant pediatric brain tumors and demonstrate how a targeted therapeutic approach is applicable in the central nervous system context to overcome this mode of immunologic resistance. Understanding how pediatric brain tumors evade the immune system can guide the development of immunotherapies, offering new hope for improved outcomes and quality of life for young patients facing these challenging conditions.

**Abstract:**

Background: The application of immunotherapy for pediatric CNS malignancies has been limited by the poorly understood immune landscape in this context. The aim of this study was to uncover the mechanisms of immune suppression common among pediatric brain tumors. Methods: We apply an immunologic clustering algorithm validated by The Cancer Genome Atlas Project to an independent pediatric CNS transcriptomic dataset. Within the clusters, the mechanisms of immunosuppression are explored via tumor microenvironment deconvolution and survival analyses to identify relevant immunosuppressive genes with translational relevance. Results: High-grade diseases fall predominantly within an immunosuppressive subtype (C4) that independently lowers overall survival time and where common immune checkpoints (e.g., PDL1, CTLA4) are less relevant. Instead, we identify several alternative immunomodulatory targets with relevance across histologic diseases. Specifically, we show how the mechanism of EZH2 inhibition to enhance tumor immunogenicity in vitro via the upregulation of MHC class 1 is applicable to a pediatric CNS oncologic context. Meanwhile, we identify that the C3 (inflammatory) immune subtype is more common in low-grade diseases and find that immune checkpoint inhibition may be an effective way to curb progression for this subset. Conclusions: Three predominant immunologic clusters are identified across pediatric brain tumors. Among high-risk diseases, the predominant immune cluster is associated with recurrent immunomodulatory genes that influence immune infiltrate, including a subset that impacts survival across histologies.

## 1. Introduction

Pediatric central nervous system (pCNS) malignancies account for most cancer mortalities among children and adolescents [1]. Treatment with surgery, radiation, and chemotherapy remains palliative in many circumstances, and survivors often experience adverse late effects. Recognizing that immune evasion is critical for cancer growth has led to novel immunotherapies designed to rescue immune activity against tumors [2,3]. Checkpoint inhibition targeting the CTLA4 and PD1/PDL1 axes has led to remarkable successes in the treatment of adult oncologic diseases [4]. Importantly, successful anti-tumor immune activity has been observed across diverse cancers treated with these therapies. Nonetheless, only a subset of patients benefits [5]. The application of immune checkpoint inhibition for childhood brain tumors has been limited, except in rare cases of a high tumor mutational burden in the setting of mismatch repair deficiency syndromes [6]. These observations suggest diverse mechanisms of immune quiescence in both adult tumors and pediatric brain tumors.

Previously, The Cancer Genome Atlas (TCGA) project utilized an immunologic gene panel to identify pan-cancer immune subtypes that convey information on immune infiltrate, immune–cancer-signaling interactions, and transcription factors that modulate the immune response [7]. Across 30 adult cancer types, this gene set resulted in samples across disease types clustering into six predominant immune subtypes, C1–C6. As summarized by Thorrson, et al., C1 was associated with angiogenic genes and a Th2 bias; C2 exhibited high M1/M2 macrophage polarization and a high CD8 signal, along with a rich T-cell receptor repertoire; C3 was enriched for Th17 and Th1 genes and lower tumor proliferation; C4 was characterized by a high M2 macrophage signature and Th1 repression; C5 demonstrated significantly low lymphocyte and high M2 macrophage signatures and was found mostly in low-grade gliomas; and C6 was the smallest cluster and demonstrated high lymphocyte infiltrate and a TGF-β signature [7]. Each subtype differed in leukocyte infiltration, tumor microenvironment composition, regulatory networks, and patient survival associations, with C2 and C3 associated with favorable prognosis and C4 and C6 associated with worse prognosis. This suggested that immune cancer interactions differ based on the tumor’s immune context, opening the possibility of personalized immunotherapy approaches [7].

In the meantime, the molecular heterogeneity of pediatric brain tumors has become well appreciated, as reflected in recent updates to the World Health Organization classification [8]. However, understanding the microenvironmental impacts and the mechanisms of immunosuppression in pCNS malignancies remains unresolved, hampering the development of effective immunotherapies. Recently, the Children’s Brain Tumor Network (CBTN) released gene expression profiles for ~700 primary pediatric brain tumors encompassing >30 histologic diagnoses. The database captures important clinical outcomes, including overall survival (OS), progression-free survival (PFS), extent of surgical resection, and whether a tumor sample is from a primary or recurrent/progressive presentation. We apply the TCGA immune-subtype clustering algorithm to explore how pCNS neoplasms cluster into different immune subtypes and hypothesize that high-grade pCNS malignancies share common immunosuppressive features. Although the CBTN dataset is one of the largest pCNS disease collections, if analyzed by molecular subtypes, the sample size is still presently insufficient. Therefore, our analysis focuses more broadly on histologic grade for comparison purposes, with the aim of identifying immunotherapeutic targets with broad applicability.

## 2. Materials and Methods

### 2.1. Description of Data Source

The Children’s Brain Tumor Network (CBTN) released the bulk transcriptomic profile for 906 independent brain tumor samples, encompassing cohorts from CBTN (*N* = 873) and the Pediatric Neuro-Oncology Consortium (PNOC), *N* = 33. Three non-tumor codes (arteriovenous malformation, choroid plexus cyst, and atypical lymphoid aggregate) and fifteen bone/peripheral nervous system codes (juvenile xanthogranuloma, plexiform neurofibroma, ossifying fibroma, and osteoblastoma) were excluded from analyses. For analysis purposes, the tumor grade was dichotomously characterized as low- or high-grade based on the ICD-O-3 primary brain and CNS site/histology listing, utilizing the 5th digit behavior code of 3 for high grade, except for spinal ependymoma, which was classified as low grade per its updated classification [8]. Abundant low-grade lesions in the cohort included low-grade glioma (LGG), ganglioglioma, craniopharyngioma, dysplastic neuroepithelial tumor (DNET), choroid plexus papilloma (CPP), and meningioma. Abundant high-grade lesions included medulloblastoma, high-grade glioma (HGG), ependymoma, diffuse midline glioma (DMG), atypical teratoid rhabdoid tumor (ATRT), and other embryonal tumors. Miscellaneous neoplasms with small sample sizes were grouped as other low-grade (*N* = 72) or other high-grade lesions (*N* = 32).

### 2.2. Data Availability

Feature counts, both raw and TPM/FPKM-normalized, and clinical phenotype data were downloaded from the open pediatric brain tumor atlas project, version 21 (https://github.com/AlexsLemonade/OpenPBTA-analysis), accessed on 30 September 2021.

### 2.3. Code Availability

All analyses were conducted in R studio, version 4.2.2, including model diagnostics and tests of model assumptions, and the code for reproducibility is hosted on OSF (https://osf.io/mqz4t/), along with the cleaned clinical phenotype file utilized for analysis.

False Discovery Rate (FDR) correction was used to control multiple hypothesis testing where indicated. For the Cox regression models, the assumption of proportional hazards was assessed with Schoenfeld residuals.

### 2.4. Immune Landscape Assessment of the pCNS Transcriptome

The immune subtype was predicted utilizing the package ImmuneSubtypeClassifier, version 0.1.0, which was developed and validated in TCGA adult cancers [9]. The deconvolution of immune cells via marker co-expression was performed with the package Imsig, version 1.1.3, and a correlation threshold of 0.7 [10]. This was advantageous over other published deconvolution methods, as it relies on marker co-expression within samples, resulting in superior correlation coefficients for individual cell-type deconvolution at the expense of lower resolution for immune cell subsets. The deconvolution of microglia gene signature was performed with the package BRETIGEA, version 1.0.3 [11]. A list of potentially relevant genes implicated in tumor immunomodulation was compiled from a literature review. This gene list with selected references is provided in Appendix A. Differential gene expression experiments and normalization of feature counts using the median of ratios method were performed with the package Deseq2 [12].

### 2.5. Cancer Cell Culture and T-Cell Expansion and Activation

The human medulloblastoma cell line, CHLA-01-MED (ATCC CRL-3021), was purchased from ATCC and maintained in DMEM–Nutrient Mixture F-12 (DMEM/F12) with 20 ng/mL of human recombinant EGF (Invitrogen, Carlsbad, CA, USA), 20 ng/mL of human recombinant basic FGF (Cell Sciences, Newburyport, MA, USA), and B-27 Supplement (Invitrogen, Carlsbad, CA, USA) at a final concentration of 2% (*v*/*v*). HEK-293T cells were cultured in DMEM containing 10% fetal bovine serum. All cell lines were cultivated under standard conditions, 37 °C, 5% CO_2_.

Human Peripheral Blood CD8+ T-Cells were purchased from STEMCELL Technologies (Vancouver, Canada). These cells were resuspended at 1 × 10^6^ cells/mL in freshly prepared ImmunoCult™-XF T-Cell Expansion Medium containing 10 ng/mL of recombinant human Interleukin 2 (Peprotech, Rocky Hill, NJ, USA). T-cells were activated by adding 25 uL/mL of ImmunoCult™ Human CD3/CD28/CD2 T-Cell Activator (STEMCELL Technologies, Vancouver, BC, Canada). A complete expansion medium was added to the cultures every 2 days until day 7 of culture.

### 2.6. EZH2 Knockdown

Lentiviral transfer plasmid, containing the short hairpin RNA (shRNA) oligonucleotide against *EZH2* (TRCN0000040077; target sequence: 5′-CCCAACATAGATGGACCAAAT-3′), and the corresponding empty control vector plasmid (pLKO.1-puro (SHC001)) were obtained from the MISSION^®^ shRNA library (Sigma-Aldrich, St. Louis, MO, USA). Lentiviral particles for *EZH2* knockdown and no shRNA control were produced by transfecting HEK293T cells at 80% confluence with 12.5 µg of the transfer vector and the second-generation packaging system, 2.5 µg of the envelope protein–coding plasmid pMD2.G (#12259, Addgene plasmid), and 5 µg of the packaging construct pSPAX2 (#12260, Addgene plasmid) using a standard Lipofectamine2000-mediated method. After 48 h, the virus-containing media were collected, centrifuged briefly to remove debris (500× *g* for 10 min), concentrated in Lenti-XTM Concentrator (Takara Bio USA, San Jose, CA, USA) using the manufacturer’s instructions, and stored in small aliquots at −80 °C.

CHLA-01-MED cells were seeded in 6-well plates (1 × 10^6^ cells per well) and lentiviruses, encoding either the *EZH2* shRNA or the empty vector backbone, were added at varying titers (5–30 μL), together with 3 μg/mL of polybrene. Then, 24 h later, the transduction medium was replaced with a fresh complete medium, and puromycin selection (2 μg/mL) was started. Approximately every 2–3 days, freshly prepared selection media were added to the cultures. After 2 weeks under puromycin selection, *EZH2* expression in knockdown and control cells was assessed using qPCR.

### 2.7. CD8+ T-Cell Cytotoxic Assay

To assess CD8+ T-cell-mediated cellular apoptosis in CHLA-01-MED, a fluorescent-based assay was performed. Wild-type, *EZH2* KD, and valemetostat-pretreated (1 μM for 72 h) CHLA-01-MED cells were incubated with 10 μM of CellEvent™ Caspase-3/7 Green Detection Reagent (Invitrogen, Life Technologies, Carlsbad, CA, USA) in complete media for up to 45 min. Hoechst 33342 Fluorescent Stain (Thermo Scientific, Pierce Biotechnology, Rockford, IL, USA) was then added at a concentration of 1 μM for a nuclear counterstain and incubated for 5 min. Labeling reagents and drugs were removed by centrifuging and discarding the supernatant and resuspending the cell pellet in Phenol red-free complete DMEM F12 media containing 10 ng/mL of IL2 and 2% FBS. Harvested cells were seeded into 96-well plates at a density of 50,000 cells per well. After 7 days of expansion and activation, CD8+ T-cells were stained with CellTracker™ Orange CMRA Dye (Invitrogen, Life Technologies, Carlsbad, CA, USA) and harvested by centrifuging and resuspending in Phenol red-free complete DMEM F12 media containing 10 ng/mL of IL2 and 2% FBS. Activated T-cells were then added to the wells at varying Target:Effector (Tumor:T-cell) rations (1:5, 1:10, 1:15, and 1:20). The wells were imaged using the Cytation™ Cell Imaging Multi-Mode Reader (BioTek Instruments, Winooski, VT, USA) before introducing T-cells and then every 12 h after T-cell addition. Images were captured in the DAPI, GFP, RFP, and brightfield channels. Automatic background-flattening parameters were used to remove background fluorescence from the GFP, RFP, and DAPI channels. To identify tumor cells and T-cells, primary masks were implemented in the DAPI and RFP channels, and the total cell counts were calculated using a pixel-intensity threshold of 2000. Apoptotic tumor cells were identified as a sub-fraction of the total count from DAPI using an RFP fluorescence intensity >1000 as the threshold. Two independent biological repeats of the experiment were performed, each consisting of six technical replicates per condition.

### 2.8. Western Blot Analysis

The primary antibodies, anti-EZH2 (Cell Signaling, Danvers, MA, USA), anti-GAPDH (ABclonal Technology, Woburn, MA, USA), and anti-HLA-ABC (W6/32) (Invitrogen, Carlsbad, CA, USA), were used at dilutions of 1:1000, 1:5000, and 1:500, respectively. HRP-conjugated anti-mouse (Cell Signaling, Danvers, MA, USA) and anti-rabbit (ABclonal Technology, Woburn, MA, USA) secondary antibodies were used at a 1:5000 dilution, and chemiluminescence was detected using SuperSignal™ West Pico PLUS Chemiluminescent Substrate (Thermo Fisher Scientific, Waltham, MA, USA) on an LI-COR Odyssey^®^ Fc Imaging System. Band intensities were quantified with Image Lab V5.2 Software from Bio-Rad.

### 2.9. Ethics Statement

Neither institutional review board nor ethics committee approval was required for this project, as neither animal nor human subjects were utilized.

## 3. Results

### 3.1. An Immunosuppressive Immune Subtype Is Common across High-Grade pCNS Malignancies and Correlates with Poor Survival

In the CBTN cohort, all six immune subtypes were observed across the primary and progressive/recurrent samples, although the most common subtypes were C4 (lymphocyte deplete, 62.4%), C3 (inflammatory, 18.7%), and C5 (immunologically quiet, 15.2%). C4 was the predominant immune subtype among high-grade lesions, whereas C3 was overrepresented among low-grade lesions (Figure 1A). The proportion of immune subtypes within diagnosis codes was comparable between the recurrent/progressive and primary samples.

In the adult TCGA pan-cancer cohort, the C4 immune subtype carried the worst prognosis for a given tumor type [7]. Similarly, for the CBTN cohort, a univariate Kaplan–Meier analysis showed that there was a decreased cumulative survival probability in the C4 tumors compared to all other immune subtypes (Figure 1B, log rank, *p* < 0.001). Next, we performed a multivariate proportional hazards regression analysis to assess survival after adjusting for tumor grade, extent of surgical resection, and patient age. Age was included as a surrogate for potential treatment exposures, as children under the age of 4 may have radiation omitted or delayed, whereas adolescents and young adults may not tolerate pediatric chemotherapy protocols. The results showed that after adjusting for these factors, C4 (HR 4.7, *p* = 0.003) and C5 (HR 3.5, *p* = 0.03) were associated with worse survival outcomes relative to C3 (Figure 1C).

### 3.2. Deconvolution Provides Insight into the Cellular Composition of pCNS Tumors, Stratified by Immune Subtypes

The immune system–cancer interactions, orchestrated by various components of innate and adaptive immunity, within pCNS tumor microenvironments contribute to both anti-tumor immunity and tumor immune evasion [13]. Methods to deconvolute the relative gene signatures for the immune cell type of interest from bulk tumor RNA-seq data permit insights into the tumor microenvironment and were applied to the CBTN cohort.

Previously, Nirmal, et al. developed a method (*Imsig*) to deconvolute the immune cell content from bulk RNA-seq data to provide relative abundance values of five immune cell types (T-cells, B-cells, neutrophils, NK cells, and macrophages) and two cellular processes (tumor proliferation and interferon signaling) [10]. We applied this pipeline to the CBTN cohort, *N* = 870 (Table 1), to examine the cell composition of the C3 and C4 immune subtypes. To contrast the cell composition differences between different histologic grades or between different immune subtypes, we performed logistic binomial regression analyses. The results from the first model, which compared high-grade samples to low-grade samples without regarding the immune subtype, suggest a lower NK-cell abundance (OR 0.84) and higher tumor-proliferation (OR 7.1) and interferon signatures (OR 1.4) in high-grade lesions. In the second model, the C4 immune subtype was contrasted with C3, adjusting for the effect of the tumor grade. The results suggest that C4 lesions are characterized by a lower T-cell signature (OR 0.36) and higher NK (OR 1.2), macrophage (OR 2.1), and tumor-proliferation (OR 2.9) signatures relative to C3 lesions.

To explore the impact of different immune cell compositions within different immune subtypes, we performed a Cox regression analysis between the relative abundance values for each cell type and patient survival outcomes. In high-grade tumors, we focused on overall survival (OS), whereas given the low death rate among the low-grade population, progression-free survival (PFS) was evaluated. Disease diagnosis and extent of resection were used as appropriate covariates in the analyses (Figure 2A and Appendix A). Analysis with OS in C3 high-grade tumors was not possible due to sample-size constraints (*N* = 16). Notably, increasing T-cell relative abundance was found to be associated with better PFS in C3 low-grade tumors (HR 0.6). However, it was also associated with worse PFS/OS in low- and high-grade C4 tumors (HR 1.4 and 1.3, respectively).

Surprisingly, an increasing macrophage abundance was found to be slightly protective in C4 high-grade tumors (Figure 2B, HR 0.98, *p* = 0.002). Imsig is not designed to differentiate bone marrow-derived macrophages from microglia, a unique myeloid cell in the CNS microenvironment that is embryologically distinct from bone marrow-derived macrophages and is thought to serve functions in anti-tumor immune activity [14]. Furthermore, the TCGA immune clustering does not encompass microglia. We therefore employed another deconvolution pipeline developed by McKenzie, et al. that provides microglia signatures as a surrogate proportion variable (SPV), which represents a relative abundance of microglia, for independent analysis [11,15]. Among low-grade tumors, C5 exhibited a lower microglia abundance (Figure 2B) relative to C3 and C4, whereas in high-grade tumors, C3 exhibited a higher microglia abundance compared to C4 and C5. Low-grade lesions, in general, exhibited increased microglia signatures relative to high-grade lesions (Kruskal–Wallis test, *p* < 0.01, *p* < 0.001, and *p* < 0.01 for C3, C4, and C5 comparisons, respectively). High-grade samples demonstrated median microglia SPV values, similar to samples of choroid plexus papilloma (CPP), which reside in the ventricle, and samples of cortical dysplasia, which is the best analogy to normal brain tissue in this dataset. When the relative abundance of microglia was incorporated into the model utilized for Figure 2A for high-grade tumors, after adjusting for diagnosis and extent of resection, increasing microglia was associated with improved survival, with an HR of 0.73 (0.60–0.89, *p* = 0.002). The embryonal tumors, inclusive of medulloblastoma, embryonal tumors with multilayered rosettes, and embryonal tumors with NOS demonstrated the lowest microglia signatures (Figure 2C).

### 3.3. Tumor-Driven Immunomodulation Occurs via Different Mechanisms Depending on the Immune Context

It is clear from the above analysis that there are context-dependent immunomodulatory interactions within pCNS malignancies. To explore this further, we identified the key genes and cytokines implicated in tumor immunomodulation from a literature search (Appendix A). Of note, there is no overlap between our literature-derived tumor-suppressive or anti-tumor immunomodulatory gene set and the TCGA gene set used to classify samples within immune subtypes.

Given the differential impact on outcomes by immune subtype, we were interested in the association of specific genes and cytokines implicated in tumor immunomodulation between the predominant C4 subtype and the more favorable C3 subtype. First, immunomodulatory genes and cytokines with predicted immunosuppressive functions, known to be expressed by tumor cells rather than the immune compartment, were correlated against the T-cell abundance in the C3 samples and C4 samples (Figure 3A). Among C4 tumors, a high proportion of immunosuppressive genes/cytokines (75%) exhibited a significant positive correlation with T-cell abundance, whereas relatively few positively correlated with T-cell abundance in C3 tumors (41%). Furthermore, these tumor-expressed immunosuppressive genes/cytokines generally co-expressed in C4 tumors, whereas this effect was not strongly observed in C3 tumors. Genes with negative correlations were also of interest, as these may be important for the exclusion of anti-tumor immune subsets. *HMGB1*, *KDM1A*, *EZH2*, *CD200*, and *VTCN1* co-expressed while negatively correlating with T-cell abundance in C4 tumors and correlated negatively with the major histocompatibility complex (MHC) class 1 (*HLA-A*, *HLA-B*, and *HLA-C*) (Figure 3A).

Immunomodulatory mechanisms in both the immune and tumor compartments have been targeted in various disease contexts. Targeting PDL1 and CTLA4 has led to clinically meaningful progress in adult neoplasms. However, as predicted, these genes were not upregulated in pCNS high-grade lesions when compared to samples of cortical dysplasia (Figure 3B). In contrast, among other immune checkpoints, *CD276* was strongly upregulated among the malignant diagnoses (*p* < 0.001) relative to the samples of cortical dysplasia (Figure 3B), suggesting alternative immunologic axes are more relevant to pCNS malignancies. To identify survival-related immunomodulatory genes in this cohort of pCNS neoplasms, all immunomodulator genes regardless of the predicted effect were assessed in a proportional hazard model, with an FDR < 0.05 for multiple testing correction thresholds (Figure 3C). Among all C4 tumors, *HMGB1*, *CD276*, *KDM1A*, *EZH2*, *NOX4*, *TBX21*, *EDNRB*, and *VTCN1* were significantly associated with decreased OS (HR > 1.0). *HMGB1*, *KDM1A*, *EZH2*, and *VTCN1* expression also correlated against T-cell abundance and HLA class 1 expression (Figure 3A). *CD276* and *VTCN1* functioned as immune checkpoints, and *EDNRB* impacted leukocyte adhesion and homing (Appendix A).

For C3 tumors, at the FDR < 0.05 threshold, no genes were found to be associated with PFS. This is most likely due to decreased power from a small sample size with progression events (*N* = 23 among 90 samples with PFS documentation). While nothing was significant in the univariate C3 tumor analysis, *CTLA4* trended toward an increased hazard ratio (HR 1.3, 0.97–1.9), along with *EZH2* (HR 2.0, 1.0–4.1). To further analyze C3 tumors, a stepwise regression model was performed to account for the extent of surgical resection and gene-correlation effects. In this model, incorporating genes with a univariate p-value cutoff of 0.1, both *CTLA4* (HR 1.97, *p* = 0.001) and *ADORA2A* (HR 1.85, *p* = 0.02) were associated with lower PFS, whereas *KLRK1* (HR 0.60, *p* = 0.003) was associated with higher PFS.

Taken together, varying patterns of tumor immunomodulation were observed, with a stronger immunosuppressive gene association seen in the C4 subtype. This suggests different mechanisms of immune resistance depending on the immune subtype, with translational relevance.

### 3.4. Decreased Expression of Antigen-Presenting Machinery Is Another Mechanism for Immunosuppression Observed in pCNS Malignancies and Is Modifiable Using Epigenetic Therapy

Recognition of antigens is a necessary step for T-cell-mediated anti-tumor immunity. While pCNS malignancies are often characterized by a low tumor mutational burden, it has been well documented that neoantigens resulting from events such as alternative splicing may induce an anti-tumor immune response [16]. Utilizing the CBTN cohort, we examined the expression levels of class 1 and 2 MHC genes by immune subtype (Figure 4A). The C5 subtype exhibited overall low MHC expression in both high- and low-grade lesions. For C3 and C4, low-grade diagnoses exhibited a higher MHC expression profile compared to high-grade diagnoses with the same immune subtype. This suggests that the suppression of antigen presentation represents an additional mechanism of immunosuppression in malignant pediatric brain tumors.

The regulation of MHC expression is complex but can be influenced by epigenetic modifiers [17]. *EZH2* and *KDM1A* are immune regulators that have both been implicated in silencing MHC expression [17,18]. Both transcription factors were strongly upregulated across high-grade lesions in the CBTN cohort relative to the samples of cortical dysplasia (Figure 4B). Furthermore, both correlated against MHC expression in this cohort (Figure 3A) and impacted survival parameters in the context of the predominant immune subtypes (Figure 3C), with increasing expression of *EZH2* and *KDM1A,* leading to decreased OS in the context of the C4 immune subtype.

Given the unmet need in many of these pCNS malignancies, we examined the impact of EZH2 inhibition on modifying MHC expression in the pCNS disease context, given its strong relevance as a translational target and availability of clinically utilized agents. We utilized a human medulloblastoma cell line (CHLA-01-MED), known to exhibit high expression of *EZH2* and knockdown *EZH2*, through shRNA or performed pharmacologic inhibition with valemetostat, which inhibits EZH2 and its homolog EZH1. Activated donor T-cells were co-cultured with CHLA-01-MED cells at an optimized T-cell:effector cell ratio (10:1). Within 6 h of mixing, T-cells migrated toward the tumor neuro-spheres. The T-cells could be seen invading the EZH2-inhibited neuro-spheres and breaking them apart into diffuse, cloud-like formations at 24 h (Figure 4C, middle and right upper panels). Additionally, Caspase-3/7 fluorescence, a marker for apoptosis, was also assessed at 24 h (Figure 4C lower panels) and over time (Figure 4D). While the extent of apoptosis increased with the incubation time for all conditions, the increase was significantly higher in the EZH2 knockdown and drug-treated cells (both compared to wild-type control, t-test *p* < 0.001) at time points from 12 h to 48 h. Thus, the suppression of *EZH2* in the CHLA-MED-01 model led to improved tumor-killing efficacy of activated T-cells. The increase in MHC expression after EZH2 inhibition was verified through Western blot (Figure 4E). These results indicate that epigenetic modification through the inhibition of EZH2 alters the expression of MHC, which subsequently allows for the improved allogeneic T-cell-mediated killing of CHLA-MED-01. This expands the mechanistic scope of EZH2 as a negative regulator of MHC expression [17] to pediatric CNS oncologic diseases.

## 4. Discussion

The application of immunotherapy for pediatric brain tumors remains a clinical challenge owing to the unique tumor microenvironment and protected niche by the blood–brain barrier [19]. Nonetheless, therapeutic modalities, including oncolytic viral therapy, tumor lysate vaccine approaches, adoptive cellular therapy, and checkpoint inhibitors, are under active investigation [19,20]. Many pediatric CNS malignancies remain essentially incurable, including posterior fossa group A (PFA) ependymoma, diffuse midline glioma, pediatric-type high-grade glioma, high-risk medulloblastoma (e.g., P53 altered SHH medulloblastoma), and others. Here, the development of effective immunotherapy as an alternative treatment paradigm would confer great benefits. Conversely, “benign” lesions such as low-grade gliomas and craniopharyngiomas have a strong propensity for recurrence and progression over time, and the tumor’s location and current treatments can cause morbidity, including vision loss, hypopituitarism, hypothalamic dysfunction, and specific neurologic deficits. The development of effective immunotherapy approaches for these diseases could help prolong progression-free survival and further reduce the use of radiotherapy and its late sequelae [21].

Clinical translation challenges encountered in pediatric neuro-oncology relate to the rarity of individual brain tumors and their subtypes. Often, early-phase trials are limited by a heterogeneously treated patient population and small sample sizes, despite successful infrastructure to support multi-institutional clinical investigations. In this analysis, we observed recurrent and dominant immune gene clusters across primary pediatric brain tumors in a large cohort of histologically diverse tumors with transcriptomic data. We were therefore motivated to identify potential immunologic targets that cut across histologic subtypes within the common immune subtypes. In our analysis, the lymphocyte-depleted (C4) subtype was predominant among high-grade lesions and the inflammatory (C3) subtype was more common in low-grade diseases. The lymphocyte-depleted (C4) immunologic gene cluster was also most abundant among CNS tumors in the adult TCGA cohort [7], which hints at a common immunologic milieu that supports neoplastic pathogenesis in this niche. This pan-pCNS immuno-classification was used to query differences in the immune content, survival outcomes, and immunomodulatory gene expression to identify targets with translational potential.

The mechanisms for tumor immune suppression, in general, are numerous and include deficits in the recruitment of antigen-presenting cells and antigen presentation, the exclusion of anti-tumor immune cells by the tumor microenvironment and cancer–immune signaling, and the dynamic production of chemokines/cytokines that promote tumorigenesis [22,23,24]. Within the C4 immune subtype, we found a lower T-cell abundance compared to C3. In C3, increasing T-cell abundance was associated with better PFS, whereas increasing T-cell abundance conferred worse survival in C4. Notably, similar observations were previously seen in the deconvolution of the methylation signatures of medulloblastoma and rhabdoid tumors [13]. Pro-tumorigenic T-cell populations in immune-cold settings have been characterized previously, and the mechanisms of dysfunction differ from exhausted anti-tumor immunity in the context of immune-hot cancer types [25]. We identified the correlation of multiple immunosuppressive genes and cytokines (Figure 3, Appendix A) with T-cells in the C4 subtype, and a subset additionally associated with worse outcomes among pediatric CNS neoplasms, including the transcription factors *KDM1A* (LSD1), *NOX4*, *HMGB1*, *EZH2*, and *TBX21*; the immune checkpoint *CD276* (B7-H3) and *VTCN1* (B7-H4); and Endothelin B Receptor (*EDNRB*). The finding of broadly upregulated *CD276* (Figure 3B) in this cohort and immunosuppressive cluster aligns with other reports [26], and the confirmation of its impact on survival in this large independent dataset warrants investigation into the effect of pharmacologic inhibition in high-risk pediatric CNS malignancies. Alternatively, the finding of immunomodulatory genes produced by C4 high-grade neoplasms that correlate with T-cell dysfunction (Figure 3A) may lead to strategies to improve the functionality of cellular therapy products that anchor to CD276 within the local microenvironment.

In addition to the suggestion of a dysfunctional T-cell compartment, there was a clear decrease in antigen-presenting machinery among the pCNS high-grade malignancies. This observation provides a possible explanation for why microglia, an important aspect of anti-tumor regulation involved in antigen processing [14], were found in lower abundance in these lesions. The downregulation of antigen presentation is an important mechanism of immune evasion and represents a key factor in immune checkpoint resistance [27,28]. Among C4 tumors, the transcription factors *HMGB1*, *KDM1A*, and *EZH2*, and the immune checkpoints *CD200* and *VTCN1* expression each correlated against HLA class I expression. *EZH2* is the catalytic component of the polycomb repressive complex 2 (PRC2), which serves as an epigenetic regulator involved in the methylation of histone H3 [29]. Tumors with histone 3.3 alterations, including H3K27M or H3G34 mutations and EZHIP overexpression, result in aberrant PRC2 activity [30,31]. These tumors still depend on EZH2 function and, therefore, represent a candidate target in CNS malignancies [31,32]. Furthermore, the PRC2 complex has been previously shown to result in the negative regulation of antigen presentation [17], and given the strong expression of *EZH2* in this cohort of high-grade pediatric CNS malignancies, we are interested in its potential role as an immunotherapy target. We explored the impact of EZH2 inhibition on MHC expression and found that EZH2 inhibition also increased the expression of HLA class I in a pediatric CNS cell line overexpressing EZH2, resulting in increased allogeneic recognition and killing by activated T-cells in vitro. In the context of pediatric malignancies, there is often genomic stability with low mutational burdens, but the presence of neoantigens such as alternative transcripts [16] normally silences transposable elements, and abnormal post-translational modifications represent sources of antigens that may be used to expose to the adaptive immune system through epigenetic therapy [33,34,35,36]. Doing so may enhance the efficacy of other immunotherapy modalities, such as oncolytic virotherapy and vaccine-based approaches, which, in part, rely on the ongoing presentation of tumor-associated and tumor-specific antigens.

Ultimately, our findings of a common immunosuppressive immune subtype across pCNS neoplasms and decreased antigen presentation suggest a treatment model in which the anti-tumor immune response is facilitated through enhancing antigen presentation and immunosuppressive signaling interference to promote effector cell function within the tumor microenvironment. An illustration of this approach can be found in a recent publication, where interfering with the immunosuppressive IGF axis augmented the activity of anti-GD2 CAR-T-cells against DMG [35]. We identified immunosuppressive mechanisms that correlated with worse survival and, therefore, represent additional therapeutic targets worthy of investigation. Currently, immunoproficient models of pCNS malignancies are limited compared to immunodeficient models, but various techniques for generating models in immunocompetent hosts have been described [20]. Further development of syngeneic models that recapitulate the molecular subgroups of pediatric CNS malignancies will enable mechanistic and translational investigations into novel immunotherapeutic combinations.

With regard to low-grade tumors in this cohort, there was a subset of cases associated with an inflammatory (C3) immune subtype characterized by higher T-cell abundance that correlated with better survival outcomes. This contrasts with the original TCGA cohort, where C3 was not found in high abundance in adult-type low-grade gliomas. As the latest WHO classification of CNS tumors recognizes, there are fundamental molecular differences between pediatric and adult CNS neoplasms [8]; we can conjecture that the increased presence of the C3 immune subtype in pediatric low-grade brain tumors reflects these biological differences. The pediatric C3 tumors express HLA and have a lower tumor proliferation score, and it is possible that the fundamental molecular differences between pediatric and adult-type low-grade CNS tumors contribute to these differences in a tumor-immune context. Importantly, increasing *CTLA-4* may lead to adverse progression-free survival, suggesting that T-cell silencing is a more pertinent mechanism of immune suppression in this context. Craniopharyngioma, in particular, has a high abundance of this immune subtype, which supports previous observations of an inflammatory microenvironment among craniopharyngiomas [37]. The application of current immune checkpoints may, therefore, confer benefits in this population. At the moment, a clinical trial exploring the combination of nivolumab and DAY101 in a 1:1:1 study design is enrolling through the Pediatric Neuro-Oncology Consortium (NCT05465174).

The strengths of this analysis include the systemic evaluation of the largest transcriptomic cohort of pediatric brain tumors to date. The models employed were strengthened by the availability of important clinical covariates, in particular, the extent of surgical resection, which is missing in many transcriptomic databases. Nonetheless, the sample size for the individual molecular subgroups was small and restricted by missing data, prohibiting subgroup-specific analyses. A planned release by the Childhood Brain Tumor Network will greatly expand the sample size and may permit deeper subgroup-specific analyses. The deconvolution of immune cell subsets was found to be statistically unreliable when employing multiple established methodologies, likely due to the overall rarity of the cell types present. Therefore, deconvolution based on gene-set co-expression was utilized and was found to be more reliable at the expense of immune cell subset resolution. Complementary multi-panel flow cytometry and multi-omics analyses of fresh samples would be of great benefit, and further work to define the gene sets for intra-tumoral cell populations at the single-cell level would facilitate the deconvolution of the bulk RNA-sequencing samples that have been accumulated to date [38,39]. As mentioned, the generation of molecularly relevant syngeneic models will be important for translational investigations. In this context, spatial transcriptomics may help better define the ligand–ligand receptor interactions relevant between the immune and cancer compartments.

## 5. Conclusions

The prevalence of pediatric central nervous system (pCNS) malignancies, as the leading cause of cancer mortality among children and adolescents, necessitates the development of new, effective therapies. The success of immune checkpoint inhibition targeting CTLA4 and PD1/PDL1 in adult oncologic diseases has been remarkable, but only a subset of pCNS tumors benefits, suggesting diverse mechanisms of immune quiescence in pediatric brain tumors. This study explores the immunosuppressive features of pediatric brain tumors by applying an immune-subtype clustering algorithm to the gene expression profiles of over 700 primary pediatric brain tumors. Recurrent and dominant immune gene clusters are identified across histology types, and immunologic targets for these dominant subgroups are derived. We expand the utility of epigenetic therapy to upregulate major histocompatibility complex antigen-presenting machinery in pediatric CNS diseases with a predominant immunologic signature. Deciphering the mechanisms of immune suppression by pediatric CNS malignancies will be important in the development of new treatment strategies for this population.

## Figures and Tables

**Figure 1 cancers-15-05455-f001:**
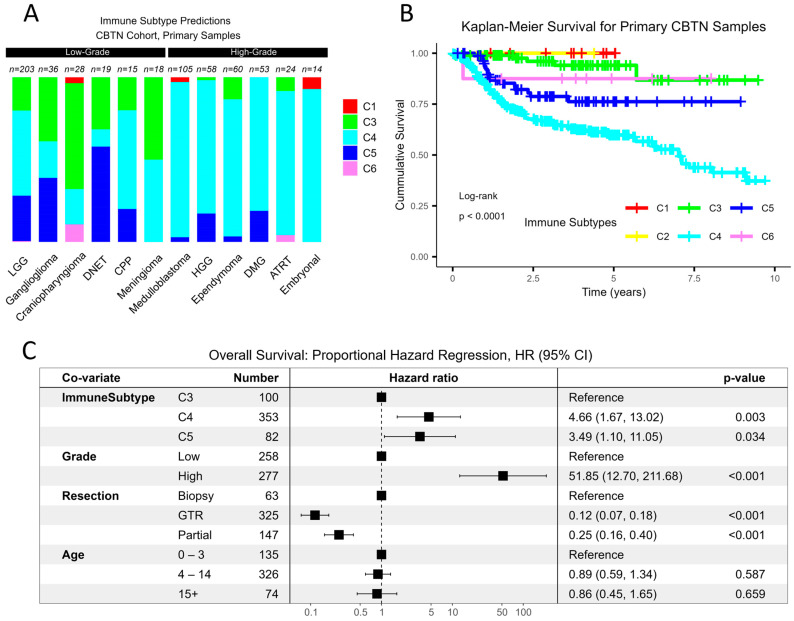
Immunosuppressive microenvironment characterizes high-grade pediatric CNS malignancies and correlates with poor survival. (**A**) For abundant diagnosis types in the CBTN cohort, the frequency of immune subtypes stratified by diagnosis is shown for the primary samples, coded as per the TCGA classification. The distribution of immune subtypes differs between high- and low-grade lesions (X^2^ = 129.1, *p*-value < 1–16), and residuals reveal a higher contribution of C4 among high-grade tumors and a higher contribution of C3 among low-grade tumors. (**B**) A univariate Kaplan–Meier curve correlates the C4 immune subtype with the worst prognosis, whereas C3 demonstrates a favorable prognosis (log rank, *p* < 0.001). (**C**) A multivariate Cox regression model was fitted to determine the immune subtype effect on survival after adjusting for tumor grade, extent of neurosurgical resection, and age. Relative to C3, both C4 (4.66, *p* = 0.003) and C5 (3.49, *p* = 0.034) subtypes exhibit an increased hazard ratio (HR) for overall survival (OS).

**Figure 2 cancers-15-05455-f002:**
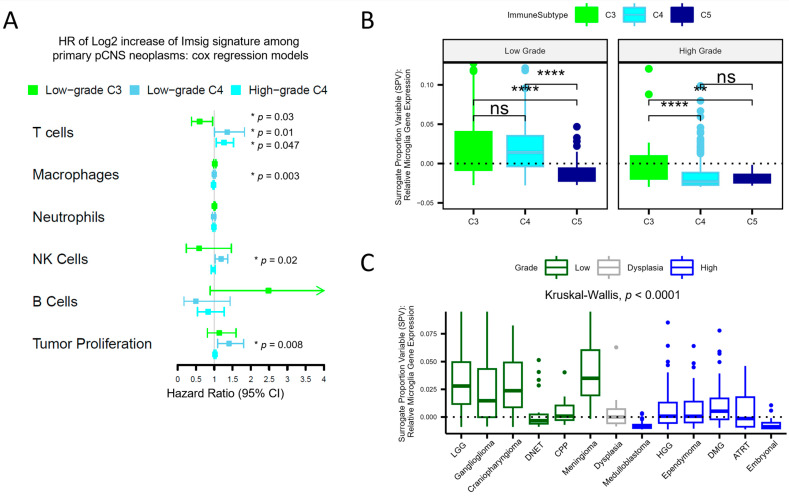
Deconvolution reveals differential immune cell infiltration based on the immune subtype. T-cell signature impacts survival parameters differentially depending on the immunologic context. (**A**) Three Cox regression models were employed to investigate the impact of immune cell abundance (Table 1) on survival parameters. The high-grade C4 model is adjusted for resection extent and diagnosis code, and the low-grade models are adjusted for resection extent. For low-grade tumors, T-cell abundance in C3 exhibits improved PFS (HR 0.60, *p* = 0.03), whereas for the C4 subtype, increasing T-cell abundance is associated with worse OS (HR 1.3, *p* = 0.013) and worse PFS (HR 1.4, *p* = 0.047) in high-grade and low-grade tumors, respectively. Appendix A displays the full model results. (**B**,**C**) Information on microglia, unique to the CNS, is not captured by the immune subtype classification. Here, the relative abundance of microglia gene signatures, expressed as a surrogate proportion variable, is shown stratified by immune subtype (**B**) and diagnosis (**C**). In low-grade tumors, C3 and C4 exhibit comparable microglia abundance, whereas that of C5 is lower; high-grade C3 demonstrates higher microglia abundance compared to high-grade C4 and C5 (Kruskal–Wallis H test, where * = *p* < 0.05, ** = *p* < 0.01, **** = *p* < 0.0001, ns = not significant). High-grade tumors exhibit comparable microglia abundance in samples of cortical dysplasia, whereas low-grade tumors exhibit increased microglia abundance (ANOVA Tukey HSD *p* adj = 0.97 and <0.001, respectively; dashed lines represent median SPV values for dysplasia).

**Figure 3 cancers-15-05455-f003:**
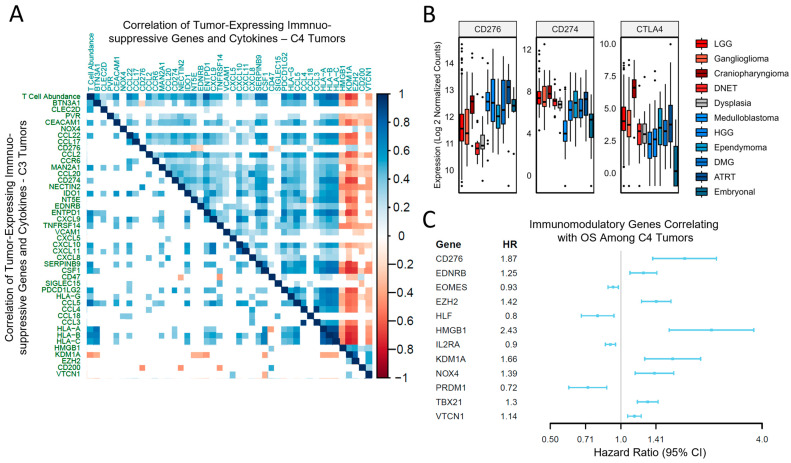
Mechanisms of immunosuppression differ depending on the immune subtype, with many immunomodulatory genes impacting survival parameters across tumor types. (**A**) Spearman-based correlation analysis was performed to identify tumor-produced immunosuppressive genes and cytokines (Appendix A) that correlate with increasing T-cell abundance, and genes for MHC class 1 (HLA-A, B, and C) are additionally included. Colored tiles demonstrate either positive (blue) or negative (red) significant correlations (Bonferroni-adjusted *p* < 0.05 threshold). This correlation matrix shows how the T-cell score correlates with each immune gene (first row/column), as well as the correlations for each immune gene with one another. For C4 tumors, the *Imsig* T-cell score strongly correlates with a large portion of these genes compared to C3 tumors. In general, the correlations among the immune genes in the C4 cluster are highly correlated with one another, but this pattern does not appear in C3. In C4 tumors, *HMGB1*, *KDM1A*, *EZH2*, *CD200*, and *VTCN1* exhibit a negative correlation with T-cell abundance and correlate negatively with HLA class I expression. (**B**) Relative to dysplasia, neither *PDL1/CD274* (ANOVA Tukey HSD *p* adj = 0.03 for decreased expression) nor *CTLA4* (*p* adj = 0.55) demonstrate increased expression among high-grade lesions, whereas alternative immune checkpoints, e.g., *CD276*, demonstrate enhanced expression (*p* adj < 0.001). (**C**) Individual genes with tumor immunomodulatory functions are implicated with OS among C4 tumors, including various transcription factors and immune checkpoint targets. Statistically significant genes in the univariate analysis are displayed. Increased expression of *CD276*, *EDNRB*, *EZH2*, *HMGB1*, *KDM1A*, *NOX4*, *TBX21*, and *VTCN1* lower OS among all C4 tumors.

**Figure 4 cancers-15-05455-f004:**
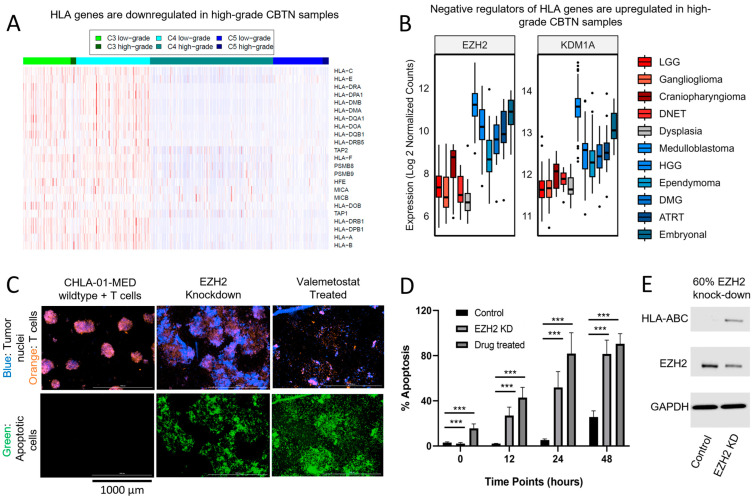
Antigen-presenting genes are downregulated in high-grade pCNS malignancies. Targeting EZH2 increases MHC, enhancing the immunogenicity of a medulloblastoma cell line (CHLA-MED-01). (**A**) Antigen presentation is critical for the development of anti-tumor immunity. Except for the C5 subtype, where antigen-presenting genes are suppressed across histologic grades, antigen suppression appears to be an additional mechanism of immunosuppression among high-grade tumors, independent of the C3/C4 dichotomy. (**B**) Two transcription factors implicated in the downregulation of MHC-related genes, *EZH2* (ANOVA Tukey HSD *p* adj < 0.001) and *KDM1A* (*p* adj < 0.001), are significantly upregulated in all high-grade lesions relative to samples of cortical dysplasia and similar expression is observed in low-grade lesions (*p* adj = 0.09 and 0.99, respectively). Sample outliers indicated by dots. In (**C**), CHLA-01-MED cells (fluorescently indicated in blue) are co-cultured with activated allogeneic donor T-cells (orange), without or with EZH2 suppression. Cells across conditions were imaged with Cytation at various time points, with a 24 h time point represented here. In the presence of EZH2 shRNA or EZH1/2 inhibition with valemetostat, an increase in Caspase-3/7 fluorescence, a marker for apoptosis (green), is observed. (**D**) Exposed conditions demonstrate an increase in Caspase-3/7 across each time point at 12, 24, and 48 h (two-sided student t-test between treated and controls, where *** = *p* < 0.001). (**E**) Activated allogeneic donor T-cells react by recognition of foreign MHC. Here, Western blot with anti-HLA-ABC is performed on wild-type and EZH2 knockdown (60% knockdown via Cytation quantification) samples, demonstrating an increase in HLA protein production in EZH2-targeted CHLA-01-MED. The uncropped bolts are shown in Appendix A.

**Table 1 cancers-15-05455-t001:** Binomial logistic regression models provide associations as odds ratios, OR (95% CI), for the relative abundance of 5 immune cell types (T-cells, B-cells, neutrophils, NK cells, and macrophages) and 2 cell processes (tumor proliferation and interferon signaling), deconvoluted from the bulk transcriptome of the CBTN cohort. The first model contrasts high-grade tumors with low-grade tumors, and the second model contrasts the C4 immune subtype with C3, with significant associations indicated in bold. C4 tumors exhibit a lower T-cell signature and higher NK-cell, macrophage, and tumor-proliferation signatures, adjusting for histologic grade.

	High Grade/Low Grade			C4/C3		
Characteristic	OR ^1^	95% CI ^1^	*p*-Value	OR^1^	95% CI ^1^	*p*-Value
T-Cells	0.69	0.43, 1.09	0.11	0.36	0.20, 0.62	**<0.001**
Neutrophils	1.01	0.81, 1.25	>0.9	0.96	0.77, 1.20	0.7
NK Cells	0.84	0.79, 0.90	**<0.001**	1.22	1.13, 1.32	**<0.001**
Macrophages	0.90	0.60, 1.34	0.6	2.05	1.28, 3.31	**0.003**
Proliferation	7.12	5.41, 9.62	**<0.001**	2.85	2.13, 3.91	**<0.001**
Interferon	1.44	1.08, 1.94	**0.013**	1.19	0.86, 1.68	0.3
Grade						
Low				—	—	
High				3.99	2.19, 7.48	**<0.001**

^1^ OR = Odds Ratio, CI = Confidence Interval.

## Data Availability

The data utilized in this analysis was provided by the Childhood Brain Tumor Networt, downloaded from the open pediatric brain tumor atlas project, version 21 (https://github.com/AlexsLemonade/OpenPBTA-analysis), accessed on 30 September 2021.

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
