# Peer review of "Revealing Pan-Histology Immunomodulatory Targets in Pediatric Central Nervous System Tumors"

_cancers, 2023, doi:10.3390/cancers15225455_

Round 1

Reviewer 1 Report

Comments and Suggestions for Authors

The article presented by Robert T. Galvin et al. is a well-written work with a comprehensive evaluation of the immune infiltration of pediatric central nervous system (pCNS) malignancies. The goal is to identify and establish an immune classification system for different types of pediatric brain malignancies, aiming to investigate differences in immune content, survival outcomes, and the expression of immunomodulatory genes to identify potential translational targets.

The results obtained from this computational analysis using publicly available databases are truly remarkable, and when combined with the in vitro assay results, they provide a clear example of the impact of this type of analysis. I believe the article is well-structured, with a robust discussion that merits its publication in the prestigious journal Cancers.

Author Response

Response: Thank you very much for taking the time to review this manuscript. We appreciate the opportunity to receive peer review and are grateful for your supportive words.

Reviewer 2 Report

Comments and Suggestions for Authors

In the manuscript titled "Revealing Pan-Histology Immunomodulatory Targets in Pediatric Central Nervous System Tumors " prepared by Galvin et al., The authors collected and analyzed a sufficiently large number of samples to explore the mechanisms of immunosuppression in childhood CNS malignancies, focusing their analysis on the C3 and C4 immune subtypes, and found that the C3 immune subtype is common in low-grade disease, and that immune checkpoint inhibitors may be more effective in this type of tumor, whereas the C4 immune subtype is common at high grades, and that it is harder for common immune checkpoint inhibitors to be effective, and in addition, the authors discovered a mechanism by which EZH2 regulates immunosuppression. Overall, the study has important implications for the discovery of different potential targets for pediatric CNS malignancies. I think the article is generally well-prepared. I have a couple of concerns as follow:

This article has done a lot of analysis and research around the immune subtypes, in order to facilitate the understanding of readers who are not familiar with this part of the content, it is recommended to add the definition or characterization of C1-C6 immune subtypes.

The authors mentioned T-cell abundance several times in the article, due to the different phenotypes of T-cells in different immune subtypes caused the opposite correlation between T-cell abundance and prognosis in different immune subtypes, it is intriguing to see what phenotypes of T-cells are related to the immune subtypes or the grading of the tumors, and it is also worthwhile to analyze whether there is a potential target of immune suppression on the T-cells.

The picture in Figure 4c is of poor quality. According to Figure 4d, the drug-intervention group obtained better killing results, while it looks like the knockout group had better results than the drug-treatment group in Figure 4c, so the scale needs to be labeled clearly.

It is well known that the C4 subtype of tumors is characterized by lymphocyte exhaustion, and it is also mentioned in the article that highly graded tumors are mostly dominated by C4, which is more difficult to treat relative to other types compared to the purity of the tumor. The authors found that CD276 expression was elevated in C4 high-grade tumors, so it is intriguing to see if CD276 inhibitors are more efficacious relative to other immune checkpoint inhibitors.

Comments on the Quality of English Language

Moderate editing of English language is required.

Author Response

Response: Thank you very much for taking the time to review this manuscript and for providing helpful feedback that will strengthen the manuscript.

The original TCGA publication describing the immune subtypes is now briefly summarized in the introduction (lines 49 – 59), and we hope this will provide further information for readers and prompt a closer look at the source document.

Your question regarding differences between T-cell subtypes between groups is an important one, and one that we struggled to investigate with the current database. Unfortunately, it is known that the overall immune infiltrate in pediatric brain tumors, especially malignant lesions, is low. Consistent with this, we found that attempts to deconvolute the signatures of T-cell subtypes was statistically unreliable with multiple methodologies employed. Given the source data is bulk RNA-sequencing, deconvolution of rarer and rarer cell populations will always be unreliable. This is highlighted as a weakness in our discussion, and we proposed possible solutions in future studies (lines 553 – 560). Because of these challenges, we instead focused on functional survival analyses as a surrogate for T-cell phenotype (e.g. figure 2A). Both our analysis and the original TCGA cohort identified C3 tumors as correlating with better prognosis, and the C3 subtype was found to be enriched for Th1 and Th17 T-cells in the larger TCGA cohort. Along the same lines, the TCGA cohort identified a high M2 macrophage signature and Th1 T-cell repression in the C4 subtype, consistent with our functional analysis. Additionally, we find that the C4 subtype is much more enriched for immunosuppressive genes consistent with an immunosuppressive tumor environment. We therefore conclude in our manuscript that both the immune abundance and the immune phenotypes differ between groups in pediatric brain tumors, even if we cannot deconvolute the specific cell types.

Thank you for your diligence in assessing figure 4. We can see that the image we choose in figure 4C does not visually match the quantification performed for figure 4D but can affirm that the quantification is accurate. Part of what is going on is that the drug-treated cells were overall more diffuse in the plate, while the wildtype and EZH2 knock-down cells were more spheroid / compact. Therefore, the well may show less focal fluorescence but overall have higher fluorescence quantitatively. However, this experiment was performed with 6 replicates, and the experiment was repeated twice so there are many representative images to choose from. For your reference, we changed the image used for figure 4c. None of these changes affect the take-home message that the T-cells were more effective at inducing apoptosis after EZH2 suppression. With regards to the image quality, this should be improved upon production if the manuscript is accepted – the images are natively high resolution but were compressed to fit onto a word document. With our submission, we include high-resolution figures for the journal to process. For your reference, we adjusted the size of the panels to make Figure 4C larger and included a scale for the photographs on the figure.

Finally, for your final comment, we agree that there are other targets worthwhile exploring for pediatric immunotherapy. We hope readers find that this analysis has pinpointed a short list that correlates with functional outcomes, including CD276. We do discuss the role of targeting CD276 in lines 482 – 488.

Reviewer 3 Report

Comments and Suggestions for Authors cancers
Article
Revealing Pan-Histology Immunomodulatory Targets in Pediatric Central Nervous System Tumors

Title is ok,
Authors have taken up a common problem to find a solution for CNS systems..

Figures were adequate, tables are ok, has this work been attempted previously?
 The literature review was done but not adequate, please check and if possible add a few more.
 English can be improved to a large extent.

To summarize, three predominant immunologic clusters are identified across pediatric brain tumors.
Among high-risk diseases, the predominant immune cluster is associated with recurrent immuno-
modulatory genes that influence immune infiltration, including a subset that impacts survival across histologies.

Accept it for publication with minor mandatory changes.

With Regards,

Author Response

Response: Thank you very much for taking the time to review this manuscript and for providing helpful feedback that will strengthen the manuscript.

In regards to your first question, the bulk of molecular profiling historically for pediatric brain tumors has been based on methylation arrays. Grabovska, et al reported immune clusters derived from the methylome of three tumor types and used methylCIBERSORT to perform immune deconvolution. Petralia, et al performed proteomic characterizations on 7 histologies encompassing 6,429 proteins. This work focused more on biological investigations, but their immune clustering was similar to our findings but more limited in scope. Our study contributes to this prior work by analyzing the transcriptome of a more diverse and larger cohort of pediatric brain tumors, utilized gene co-expression networks to improve the deconvolution of tumor cell subsets, and importantly performed functional analyses on a broad array of immunomodulatory genes with translational relevance. Other works focus on singular disease entities, which have tremendous value. The approach we took was to identify common themes across the various diagnoses based on transcriptomic immunologic clusters. A true barrier to improving clinical care for children with brain tumors is conducting molecular subtype specific clinical trials, as each disease and each molecular subtype of a disease are generally very rare, hindering clinical trial recruitment. By identifying common immunologic clusters and mechanisms within clusters, we aim to identify targets that may be broadly applicable.

Each of these studies have been referenced and discussed in the manuscript. We additional provide >100 more references within supplementary table 2 that could not fit into the main body of the manuscript. We have now included additional references reviewing the state of the field and discussing the future direction of single cell sequencing efforts for pediatric brain tumors.

Finally, we have gone through the manuscript and fixed any remaining grammatical and/or typographical errors.

Round 2

Reviewer 2 Report

Comments and Suggestions for Authors

The revised version addressed my comments. Some other minor midifications are required as follows:

1.Figure 2A, 3A & 4A: delete the red underlines.

2. Figure 4C: sacle bar could be 1mm and embeded in the photo. Also please delete the original thin and faint sacle bars in the original photos.

3. Please swith the positions of conclusion and discussion sections in the main texts.